# Pathogen Moonlighting Proteins: From Ancestral Key Metabolic Enzymes to Virulence Factors

**DOI:** 10.3390/microorganisms9061300

**Published:** 2021-06-15

**Authors:** Luis Franco-Serrano, David Sánchez-Redondo, Araceli Nájar-García, Sergio Hernández, Isaac Amela, Josep Antoni Perez-Pons, Jaume Piñol, Angel Mozo-Villarias, Juan Cedano, Enrique Querol

**Affiliations:** Institut de Biotecnologia i Biomedicina and Departament de Bioquímica i Biologia Molecular, Universitat Autònoma de Barcelona, Cerdanyola del Vallés, 08193 Barcelona, Spain; luis.franco@uab.cat (L.F.-S.); davsanred@uoc.edu (D.S.-R.); aracelinajargarcia@gmail.com (A.N.-G.); sranzani@gmail.com (S.H.); iamela@bioinf.uab.cat (I.A.); josepantoni.perez@uab.cat (J.A.P.-P.); Jaume.pinyol@uab.cat (J.P.); angel.mozo@mex.udl.cat (A.M.-V.); JuanAntonio.Cedano@uab.cat (J.C.)

**Keywords:** moonlighting proteins, multitasking proteins, microbial pathogens, pathogen virulence factors, vaccines

## Abstract

Moonlighting and multitasking proteins refer to proteins with two or more functions performed by a single polypeptide chain. An amazing example of the Gain of Function (GoF) phenomenon of these proteins is that 25% of the moonlighting functions of our Multitasking Proteins Database (MultitaskProtDB-II) are related to pathogen virulence activity. Moreover, they usually have a canonical function belonging to highly conserved ancestral key functions, and their moonlighting functions are often involved in inducing extracellular matrix (ECM) protein remodeling. There are three main questions in the context of moonlighting proteins in pathogen virulence: (A) Why are a high percentage of pathogen moonlighting proteins involved in virulence? (B) Why do most of the canonical functions of these moonlighting proteins belong to primary metabolism? Moreover, why are they common in many pathogen species? (C) How are these different protein sequences and structures able to bind the same set of host ECM protein targets, mainly plasminogen (PLG), and colonize host tissues? By means of an extensive bioinformatics analysis, we suggest answers and approaches to these questions. There are three main ideas derived from the work: first, moonlighting proteins are not good candidates for vaccines. Second, several motifs that might be important in the adhesion to the ECM were identified. Third, an overrepresentation of GO codes related with virulence in moonlighting proteins were seen.

## 1. Introduction

Moonlighting proteins refer to those proteins that present alternative functions (named canonical and moonlighting) performed by a single polypeptide chain. They can be monomeric or multimeric. Multitasking is mostly affected by cellular localization, cell type, oligomeric state, concentration of cellular ligands, substrates, cofactors, products, or post-translational modifications [1,2,3,4,5]. Usually, moonlighting proteins are experimentally revealed by serendipity.

There are two main multitasking/moonlighting protein databases: MultitaskProtDB-II [6] and MoonProt [7]. It is remarkable that 25% of the total reported moonlighting proteins—about 1000—correspond to those involved in pathogen virulence. Appendix A shows an updated list of 258 moonlighting proteins related to pathogen virulence. As can be seen, their canonical functions correspond to ancestral and highly conserved key biological functions, such as glycolysis, Krebs cycle, basic pathways, etc. Most or all of these enzymes present moonlighting functions—though not always related to pathogen virulence. Henderson and Martin conducted an excellent review of these virulence proteins [8]. The secretion mechanism of cytoplasmatic moonlighting proteins is not well understood yet. Most pathogen moonlighting proteins are intracellular and lack a canonical signal peptide or secretion motifs. However, they are secreted outside the bacterial cell or attached to the cell membrane by an alternative mechanism and there they interact with host extracellular matrix (ECM) proteins [9]. Two recent works differentiate on cell wall proteomics—on whether yeast external GAPDH is a product of secretion or a cellular leakage, with nonconventional secretion being prevalent [10,11].

The most studied motif in the interaction between a moonlighting protein and its host is that of the enolase from *Streptococcus pneumoniae* and human plasminogen. A second example is that of Candida. The answer as to how different pathogen moonlighting proteins bind different protein sequences/structures of host ECM protein targets is still unknown. The main problem is the lack of solved co-crystal structures and the poor host–pathogen interactomics data present on PPIs databases. Our bioinformatics approach—disclosing shared motifs, computer modeling, etc.—can, to some extent, help provide insight into this issue.

In two previous works [12,13], we proposed that the immune system would discard eliciting protective antibodies against pathogen proteins that share epitopes with host proteins. Given that, one of the most important tasks of the immune system is the differentiation between self and non-self-antigens to avoid autoimmune diseases. This process is known as epitope mimicry and it is frequently used by pathogenic microorganisms [14]. Epitope mimicry appears when a stretch of shared sequence, called ‘‘mimetope’’, exists between a protein of a certain pathogen and a protein of its host. In some cases, this may lead to autoimmune reactions, most of them related to several diseases. The host immune system can become immunotolerant to avoid the autoimmune disease [15]. Proof of this epitope mimicry in moonlighting virulence proteins of pathogenic microorganisms is that they belong to key and highly conserved functions. Therefore, the host immune system will not elicit protective antibodies against the pathogen. In summary, a “clue” function might be to hide the pathogen from the host immune system. In a previous study attempting to design a recombinant vaccine against *Streptococcus uberis* (the agent responsible for bovine mastitis), four putative moonlighting proteins were checked as potential antigens. They showed good spots in the differential immune proteomes. However, they were not protective when tested experimentally [16]. In the present work, we extend our previous hypothesis, in order to gain insight into the questions and paradoxes that these proteins cause to the host–pathogen relationship. In this sense, there are three main questions that moonlighting proteins involved in pathogen virulence pose to us.


Why are a high percentage of pathogen moonlighting proteins involved in virulence?Why do most of the canonical functions of these moonlighting proteins belong to primary metabolism? Moreover, why are they common in many pathogen species?How are these different protein sequences and structures able to bind the same set of host ECM protein targets, mainly plasminogen (PLG), and colonize host tissues?


## 2. Materials and Methods

Pathogen virulence proteins were collected from MultitaskProtDB-II [6] at http://wallace.uab.es/multitaskII, accessed on 26 March 2021. Protein sequences were taken from the NCBI server (https://ncbi.nlm.nih.gov, accessed on 26 March 2021) and protein structures from PDB (https://www.pdb.org, accessed on 26 March 2021). Some protein characteristics were found in UniProt database (https://www.uniprot.org/, accessed on 26 March 2021). Functional semantic annotations were gathered from GO (http://www.geneontology.org, accessed on 30 May 2020). Human proteins involved in metastasis were found in the Human Cancer Metastasis Database (https://hcmdb.i-sanger.com/, accessed on 30 May 2020).

Protein sequence alignments were performed with BLASTP [17] of the NCBI server (https://blast.ncbi.nlm.nih.gov/Blast.cgi, accessed on 26 March 2021) against “non-redundant protein sequences” database and multi-alignments were done with Clustal-Omega [18] at https://www.ebi.ac.uk/Tools/msa/clustalo/, accessed on 9 June 2019. Both analyses were performed under the server default parameters.

Searches of motifs putatively related with the interaction with ECM proteins were performed with Minimotif Miner [19] at https://mnm.engr.uconn.edu, accessed on 18 February 2010 and DREME (Discriminative Regular Expression Motif Elicitation) [20] at https://meme-suite.org/meme/tools/dreme, accessed on 14 June 2019 using an E-value threshold of 0.05 and unlimited motif count. Selected motifs were those statistically significant (*p* < 0.05). The search has been performed using a script programmed in Python (https://www.python.org/, accessed on 9 June 2019). In Appendix A a flowchart of the program and the script can be found.

Interactomics partners were searched at BioGRID (https://thebiogrid.org/, accessed on 9 June 2019) and APID (http://apid.dep.usal.es, accessed on 9 June 2019) servers using the “All organisms” parameter. Structural docking between PLG and enolase was performed by ClusPro 2.0 [21] at https://cluspro.bu.edu/, accessed on 31 October 2017 and visualized by PyMOL (the PyMOL Molecular Graphics System, Version 2.0 Schrödinger, LLC at https://pymol.org/, accessed on 31 October 2017), both using the default parameters.

Continuous B-cell epitopes of the human protein orthologues of the pathogen virulence proteins (i.e., enolases) were predicted by the algorithm BepiPred [22] at http://www.cbs.dtu.dk/services/BepiPred/, accessed on 23 October 2017 using the defaults parameters. Vaccine candidate proteins were from the VIOLINet database [23], which shows the status of hundreds of putative vaccines (already on the market, licensed, in research, etc.).

The following detailed analyses were carried out with the two moonlighting protein classes most involved in virulence, enolases, and glyceraldehyde-3-phosphate dehydrogenase (GAPDH) (see Appendix A).

## 3. Results and Discussion

### 3.1. Moonlighting Proteins and Virulence: Questions 1 and 2

Virulence related moonlighting proteins represent around 25% of the moonlighting proteins listed in our database. In this work, we collected more virulence-related moonlighting proteins. The updated list is shown in Appendix A. The most common interaction partner protein of the host is PLG and other ECM proteins. This can be seen in Appendix A, where they are classified by the host partner protein. Moreover, Appendix A, shows the frequency of canonical functions of the moonlighting proteins involved in virulence and Appendix A shows virulence moonlighting proteins classified by microorganisms. The canonical function of moonlighting proteins involved in virulence is, most of the time, an intracellular enzyme involved in key pathways, such as glycolysis or Krebs cycle [8]. In two previous works [12,13], we proposed that these protein sequences/structures are highly conserved; thus, probably sharing epitopes. The host immune system should avoid eliciting protective antibodies against the pathogen, in order to elude a putative autoimmune response mediated by the elicited epitope mimicry [14]. These pathogens show large amino acids stretches shared with human GAPDH (Figure 1). Moreover, these regions tend to coincide with the BepiPred-predicted linear B-cell epitopes. In one of our previous works, many other examples of multi-alignments and epitope predictions can be found [13]. These multi-alignments show two remarkable characteristics: (a) there is a good match between the host and the pathogen proteins, both in stretches of conserved amino acid sequences and predicted epitopes, and (b) there are more than a few shared epitopes between them. It was previously reported that a unique shared epitope could give rise to an autoimmune response [12,13,14].

Many pathogenic microorganisms present a complex set of moonlighting proteins involved in virulence instead of one or few proteins [24]. This fact probably represents insurance against mutations or defense systems from the host. This may disable the protein and block the infection process if the virulence of the pathogen depends only in one or a few virulence secreted factors. It also permits a wide and better covering of the “surfome” and can expose a wide-ranging set of different epitopes that might be shared between the host and the pathogen. This should hide the pathogen from the host immune system.

These results led us to prompt a suggestion for vaccine design experiments, especially for subunit recombinant ones. Sequentially conserved moonlighting proteins are not good vaccine candidates. In a previous work [13], we conducted an exhaustive inspection of the VIOLINet database and found that proteins from successful and marketed vaccines did not have sequence homology with the host proteome. There is no successful vaccine based on a moonlighting protein. Theoretically, a successful recombinant vaccine that is designed against a certain pathogen should generate protective immune response against others, especially those from the same genera. However, this does not occur even in species phylogenetically as close as *Neisseria gonorrhoeae* and *N. meningitidis*. In fact, a strategy based on designing a non-virulent microorganism by deleting these virulence proteins will not be viable because their canonical functions are essential for the survival of the cell. Other examples are the vaccine candidate against *Actinobacillus pleuropneumoniae* that are based on its hemolysins ApxI and ApxII (patent WO2013/068629A1). These two proteins fit the criteria stated above.

If these moonlighting proteins were not responsible of key functions, deleting the gene would lead to non-virulent strains and their putative use in recombinant vaccines. Since the function of pathogen moonlighting proteins is related to key metabolism and, hence, essential proteins, it is difficult to generate mutants or full-gene knockouts in order to perform direct experimental demonstrations on their virulence involvement. Nevertheless, some newly designed and naturally occurring mutants do not lose the essential canonical function, but become non-virulent in vivo challenges. For example, in anaerobic conditions, bacteria lack an operative glycolytic pathway. *Neisseria meningitidis*, which uses fructose-1,6 biphosphate aldolase as an adhesin, loses the adherence capacity to the epithelial and endothelial cells when it is isogenically mutated, but this capacity returns in the complementation assay [25]. Another example of modification of a glycolytic enzyme without losing its canonical function is glyceraldehyde phosphate dehydrogenase of *Streptococcus pyogenes*. Here, a small hydrophobic amino acid sequence was added, reducing the amounts of GAPDH at the surface of the microorganism and this mutant loses its virulence in mice [26]. These examples suggest approaches to reduce, or fully eliminate, pathogen virulence based on moonlighting proteins.

In summary, our hypothesis is that pathogenic microorganisms secrete proteins that share amino acids sequences with their host to escape from its immune response. These proteins are classified as moonlighting. The canonical function of these proteins is determined by a certain amino acids sequence that has been conserved throughout the phylogeny. The enzymes of the primary metabolism accomplish this requirement. This hypothesis suggests an important idea, usable in recombinant vaccine design: the use of proteins whose amino acid sequences share stretches between the pathogen and the host to design a new vaccine is not recommended. As moonlighting proteins are abundant on the pathogenic surface, they might be selected as vaccine targets. We suggest a previous bioinformatics analysis on these proteins to analyze whether they share epitopes with host proteins.

### 3.2. Putative Involvement of Moonlighting Proteins in the Interaction with ECM Proteins, Tissue Colonization, and Metastasis: Question 3

Most pathogen moonlighting proteins related to virulence are intracellular (mainly from primary metabolism, gene regulation, etc.), as can be seen in Appendix A. They do not show a canonical signal peptide but, curiously, they are secreted outside the microorganism cell membrane using a not-yet very well described mechanism. SecA2 or T5SS systems, among others, seem to facilitate the secretion of internal proteins in bacteria [27]. Other mechanisms could be as simple as the lysis of bacteria during infection, which releases these proteins into the extracellular environment. Although the export mechanism is not relevant for this work, there is a clear relation between moonlighting functions and pathogenicity. For example, streptococcal surface dehydrogenase (SDH), which is actually a GAPDH, is a major surface protein of group A streptococci. This protein binds mammalian proteins, including PLG, by adding a 12-amino acid hydrophobic tail in its C-terminal end that blocks its export to the surface without compromising its enzymatic activity. The mutant is completely attenuated for virulence in a mouse model of peritonitis. In addition, the gene expression pattern was strongly altered, losing its virulent phenotype [1]. These bacterial mechanisms are reminiscent of those occurring in host cells. For example, macrophages recruit surface GAPDH, where it also functions as a PLG receptor. The PLG binding with GAPDH allows it to digest the extracellular matrix, thus facilitating macrophage migration [28]. By emulating this mechanism, bacteria are able to use host repair mechanisms to colonize tissue. In other words, these export mechanisms, although not well characterized, are part of the virulence mechanism, and could therefore be a new therapeutic target in the future.

A more challenging issue is how such structurally different pathogen proteins could bind the same set of host extracellular matrix protein targets (mainly PLG). We followed three approaches: (a) checked whether the virulence proteins shared clue interaction motifs or domains, suggesting an interaction with ECM proteins; (b) searched for protein interactomics partners of pathogen virulence proteins; and (c) modeled the interaction between PLG and some pathogen virulence proteins.

The first approach was to check whether the virulence proteins shared domains or motifs suggesting interaction with ECM proteins. The search for protein sequence motifs/domains related to moonlighting identification have been widely used [29,30,31,32,33,34], but not, as far as we know, if these mechanisms are shared between different moonlighting proteins. A problem with sequence motif/domain databases and servers (i.e., Prosite [35], Pfam [36], InterPro [37], etc.) is that they are very curated in order to optimize the identification of the canonical function in detriment of additional functions, which are supposed to be moonlighting functions [32]. For this reason, we used programs that employ shorter sequence motifs, such as Minimotif Miner [19] and DREME [20], although they can yield more false positives and must be reviewed manually. Shorter motifs correspond to amino acid sequence stretches, mainly of 3–10 residues. They are very conserved, and many times related to the function, interaction, localization, protease cleavage, or post-translational modification of the proteins.

We predicted two enolase motifs that might interact with PLG, highlighted in blue in Table 1. The already known motifs and their functional and structural roles are also shown. One of the predicted motif matches determined, quite well, the experimentally region. Both motifs are localized in the protein surface, as can be seen in Figure 2. Additionally, a detailed analysis of these motifs was carried out, searching for functions and protein–protein interactions involved in virulence. Results show the interaction of enolase with partners related to human ECM proteins, regulation of immune system, angiogenesis, etc. (Figure 3). From this analysis, it can be seen that the functions of these motifs are mainly related to amino acid domains involved in protein–protein interaction (i.e., SH2 domains, regulation of the immune response and signaling, etc.). The other predicted motif, colored in yellow in the structure, also binds a SH2 domain of protein PLC gamma 1, which is the substrate of the Heparin-binding growth factor 1 (acidic fibroblast growth factor). SH2 domains are involved in protein interaction and that is why both human proteins are closely related with PLG, blood functions, and immune regulation. From the PDB database, the most relevant secondary structures that are involved in the interaction of PLG with the enolase of *Streptococcus pneumoniae* and the M protein of *Streptococcus pyogenes* are alpha helices (Appendix A). Another reported motif of interaction between host plasminogen and pathogen is that of GAPDH from *Mycoplasma pneumoniae* [38]. It corresponds to the alpha helix C-terminal sequence—^226^QLVRVVNYCAKL^337^—. Recently, Satala et al. [39] have identified, by chemical crosslinking, a peptide—DKAGYKGKVGIAMDVASSEFYKDGK—from the *Candida albicans* enolase that binds human ECMs. Moreover, C-terminal lysine residues are suggested to be important in the plasminogen binding activity [40]. An alignment of *Streptococcus* and *Candida* enolases share a good level of amino acids sequences. Moreover, *Candida* enolase shares a part of the *Streptococcus* sequence motif FYDKERKVY, Table 1 (from amino acids 253 to 260).

The ECM is composed of proteoglycans and glycoproteins, and the output of servers, such as Minimotif Miner [19], show motifs for proteoglycan binding in moonlighting virulence proteins, such as GAPDH, enolases, etc. We performed a bioinformatics analysis to identify motifs that might be important in the adhesion to the ECM. In Appendix A, we collected the 17 most representative motifs found in the pathogen GAPDH and enolases. Moreover, in Appendix A we show the location of the identified motifs inside the sequence of the GAPDH and enolase, respectively. Appendix A shows the structural/functional role described for enolase motifs and most of them are involved in the binding.

The second approach was to check whether pathogen moonlighting proteins interact with host proteins, a fact that is found many times in interactomics databases (BioGRID [41] and APID [42]). The main problem is the scarce number of interaction experiments performed between those heterologous partners. Of the 19 enolases and 16 GADPHs that bind PLG, and are listed in our database, BioGRID does not contain any, and APID only shows an interaction between enolase P75189 of *Mycoplasma pneumoniae* and human PLG. GAPDH of *Mycoplasma suis* has two pig partners, actin and a structural protein (Appendix A).

The third approach was to perform a protein docking of the interaction among PLG, enolases, and GAPDH. The ClusPro 2.0 server [21] was used because it allows dockings of big proteins, such as PLG, and some moonlighting proteins. Of the 35 proteins that bind PLG, 14 present an already solved PDB structure. When the ClusPro 2.0 docking was used, the resulting model with the lowest energy was chosen and visualized with PyMOL. The modeling indicates that several GAPDH and enolases bind the same amino acids, some of them in the area of the motifs described in Figure 2. All details can be seen in Appendix A. For example, it is remarkable that when the two key amino acids of PLG, Arg561 and Val562, are disrupted, PLG is activated to plasmin. As can be seen in Appendix A, the docking modeling of the PLG binding sites shows that they are effectively docked using these amino acids.

The answer as to how the different pathogen moonlighting proteins bind so many different protein sequences/structures of the host ECM protein targets is still unknown. The main problem is the lack of solved co-crystal structures and the poor host–pathogen interactomics data present on PPI databases. Our bioinformatics approach—disclosing shared motifs, using computer modeling, etc.—can help provide insight into this issue. In addition, the microorganism colonization mechanism—through moonlighting proteins—shows a parallelism with cancer colonization and tissue remodeling that can help decipher the mechanism of infection.

### 3.3. Tissue Remodeling Strongly Suggests a Parallelism between Microbial Tissue Colonization and Cancer Metastasis

The host extracellular matrix (ECM) is a widely distributed proteinaceous tissue being the key component of the connective tissue and basement membranes. Adhesion to ECM proteins is, therefore, a major strategy for pathogen colonization and invasion. The above results, supported by the analyses of structural motifs, reinforce our idea of the existence of interactions between pathogen and host proteins that might be involved in tissue remodeling. In fact, it was reported elsewhere that PLG is involved in microbial infection. Interaction of microbial virulence proteins with PLG would induce a structural change of PLG that facilitates its conversion to plasmin. This should promote the degradation of the extracellular matrix and microbial colonization, as in the metastasis process [43,44,45,46,47,48].

The importance of the extracellular matrix (ECM) becomes clear when we consider that it constitutes the basic scaffold of tissues and organs. However, it is a dynamic structure that is subject to constant remodeling, in response to functional needs or tissue damage. Therefore, there are highly conserved mechanisms in all animal phyla to perform these tasks. Since the activation of these mechanisms, which allow cells of the immune system or stem cells to colonize tissue for repair, they can also be used by pathogens to invade tissue, or tumor cells to infiltrate and nest in organs. Thus, we find the same set of moonlighting proteins involved in matrix remodeling in both physiological processes and pathological ones, such as infection or cancer.

It was also reported that a number of human moonlighting proteins are involved in human cancer, angiogenesis and invasion, antiapoptotic pathways, etc. Some examples are Hsp90, transglutaminase 2, HMGB1, p53, ESE-1, or β-catenin [49,50,51,52,53]. The question is whether the pathogen moonlighting proteins involved in invasion and colonization share protein characteristics, functions, targets, and use of similar processes, such as tumor metastasis. To gain insight into this mechanism, we checked whether the pathogen moonlighting proteins share similar GO codes and partners with human proteins that are involved in cancer metastasis. Human proteins involved in metastasis were collected from HCMDB [54] and their GO codes were obtained from UniProtKB [55] by means of the Retrieve/ID Mapping Tool. Several simple Python programs were used to make a set of metastasis proteins. The program parameters were adjusted to select some key general functions descriptors related with cancer (i.e., extracellular matrix, PLG, glucose metabolism, tissue) and then to search for their shared GO codes. The shared GO codes can be found in Appendix A. Among them, there are codes of interaction with components of the ECM, cell-to-cell attachment, and protease binding, as well as some related with cytokines and modulation of the immune system. From these preliminary results, it seems plausible that pathogen colonization and invasion shares characteristics with cancer metastasis through ECM proteins. Further work using this mechanism is required. The program code can be found at Appendix A.

Some proteins, such as human alpha-enolase, have been observed on the cell surface of pancreatic, breast, and lung cancer cells. Alpha-enolase surface expression was found to be dependent on the pathophysiological conditions of the cells, expression is enhanced in some types of cancer. The C-terminal residue contains a lysine that acts as a PLG binding receptor that modulates peri-cellular fibrinolytic activity and promotes the migration and metastasis of cancer cells [56]. This C-terminal domain with lysine is also present in other virulence-linked bacterial enolases. However, PLG activation mechanisms seem to follow different pathways, some of them common to other proteins important in tissue pathophysiology. For example, the *Candida albicans* enolase binding motif with PLG _235_DKAGYKGKVGIAMDVASSEFYKDGK_259_ also interacts with vitronectin and fibronectin [31]. This may indicate that there are physiological servo control mechanisms in which moonlighting proteins can play a relevant role, such as some of the proteins that are used, such as markers of malignant progression, e.g., human gamma-enolase. This human moonlighting protein appears to fulfil a different functionality depending on its cellular location. High levels of activity or presence have been detected in endocrine pancreatic tumors, seminoma, medullary thyroid carcinoma, pheochromocytoma, neuroblastoma, or small cell lung carcinoma [56]. Although this protein lacks the typical C-terminal lysine of enolase (Appendix A), it has a high homology with the interaction region of *Candida albicans* enolase and PLG. On the one hand, this fact might explain its role in the evolution of cancer; on the other hand, it alerts us that the pathogens and host share moonlighting activation pathways. Although both organisms use the same biological target to activate the mechanism, they do so with different objectives.

We performed a bioinformatics analysis to determine what functions are shared between metastasis and virulence related proteins. For this analysis, we obtained a list of metastasis related proteins and the GO codes associated to these proteins from the Human Cancer Metastasis Database HCMDB (https://hcmdb.i-sanger.com/, accessed on 9 June 2021) [54] and GO (geneontology.org). We compared these GO codes with the ones obtained using the virulence related moonlighting proteins listed in Appendix A. From this analysis, we have seen that GO code 0002020, which is associated to “protease activity”, appears frequently in proteins related to metastasis and virulence. According to our results, the frequency in which a metastasis protein shares this function with a moonlighting protein related to virulence is 11.11%. It is 7.71% related to PLG, and 7.71%, considering the whole ECM. It is known that remodeling tissues or pathogen colonization requires this activity to degrade the basement barrier. This process may facilitate the tissue invasion in both metastasis and infection [57,58]. It can also be seen that GO codes related to virulence—PLG, laminin, collagen, and fibronectin binding—are overrepresented in moonlighting proteins (Table 2). All of these functions are related to tissue remodeling processes. According to our results, these processes are important for both metastasis and virulence. Therefore, moonlighting proteins might be involved in both pathologies. The complete results of the GO codes analysis are in Appendix A.

Many studies have established a parallel between the mechanisms necessary for tissue repair and the process by which cancer occurs. In the case of cancer, these control mechanisms imply a deregulation of cell growth or an attenuating absence of cell growth in the tissue. However, in the case of infectious diseases, this repair mechanisms balance might shift towards an exacerbation of the pathogen-induced inflammatory response. These parallels among tissue repair, infection, and cancer, help us to better understand why matrix remodeling is related to metastasis mechanisms. Tissue remodeling produces matrix digestion, to facilitate the accessibility of the immune system, which will remove the cellular residues from the injured tissue. At the same time, this phenomenon creates a new niche in which tissue progenitor cells nest to restore its functionality. Tissue that is stagnated at this stage will generate a niche for a long period facilitating the metastasis [59,60].

A screening of the interactomics database STRING (https://string-de.org, accessed on 9 June 2021) shows that, in cancer processes, human PLG interacts with enolase. According to our database, the enolases of 42 different microorganisms seem to be involved in the adhesion and in the colonization of host tissues. This fact suggests that there are some common functions between infection and metastasis. As previously mentioned, the poor interactomics data between pathogenic microorganisms and the corresponding hosts precludes, for the moment, a deeper proteomics analysis. An example of this relation between virulence and metastasis is heparin. It has been described that heparin contributes both to the treatment of cancer metastasis and some infectious diseases by the inhibition of fibronectin [61].

## 4. Conclusions

The sequence and epitope similarity between pathogen moonlighting proteins and host counterparts shield the pathogen from the host protective immune response. The following rule for subunit vaccine design can be suggested: it is not recommended to base these vaccines on sequentially conserved moonlighting proteins. Since many of them are antigenic, only a few of them would elicit a true protective immune response.

Pathogen moonlighting proteins bind to host PLG and to ECM proteins, such as vitronectin, fibronectin, etc. Several motifs that might be important in the adhesion to the ECM were identified. The lack of experimental data on the amino acids involved in the interaction between PLG and most moonlighting protein partners precludes a more detailed analysis, to gain insight into the predicted sequential motifs and their specific roles. However, computer modeling of the interaction between human PLG and two pathogen moonlighting proteins, GAPDH and enolases, allows the identification of some key motifs in both partners.

From our results, the existence of a parallelism between the human process of cancer metastasis and the microbial pathogen colonization mechanism was shown. That is why an overrepresentation of GO codes related to virulence in moonlighting proteins have been seen. This parallel could suggest, in some cases, similar pharmacological strategies—small dugs, MAbs, vaccines—to treat cancer and microbial infections.

## Figures and Tables

**Figure 1 microorganisms-09-01300-f001:**
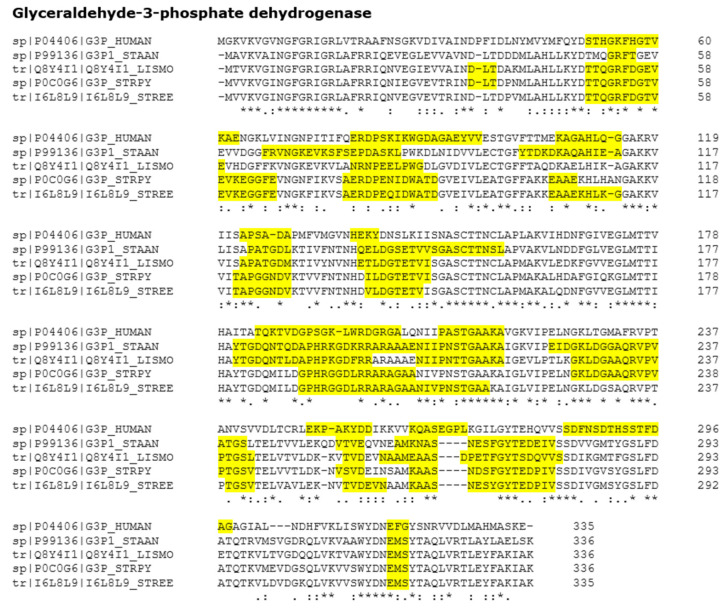
Sequence multi-alignment between human GAPDH and different pathogenic microorganisms with the predicted epitopes highlighted in yellow showing a clear conservation in the immunogenic areas of the proteins. STAAN = *Staphylococcus aureus*, LISMO = *Listeria monocytogenes,* STRPY = *Streptococcus pyogenes*, STREE = *Streptococcus pneumoniae*. Linear B-cell epitopes were predicted with BepiPred. An * (asterisk) indicates positions which have a single, fully conserved residue. A: (colon) indicates conservation between groups of strongly similar properties as below—roughly equivalent to scoring > 0.5 in the Gonnet PAM 250 matrix. A. (period) indicates conservation between groups of weakly similar properties as below—roughly equivalent to scoring =< 0.5 and > 0 in the Gonnet PAM 250 matrix.

**Figure 2 microorganisms-09-01300-f002:**
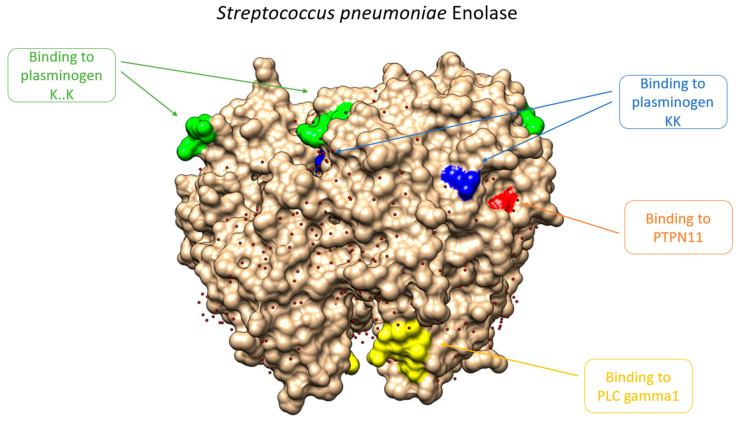
Structural view of *Streptococcus pneumoniae* enolase showing different PLG binding motifs. In green and blue are the plasminogen binding motifs described in the bibliography. In yellow and red are the bioinformatically predicted motifs that are related to virulence or immunoediting process. More information about these motifs can be founf in Table 1. W6T.

**Figure 3 microorganisms-09-01300-f003:**
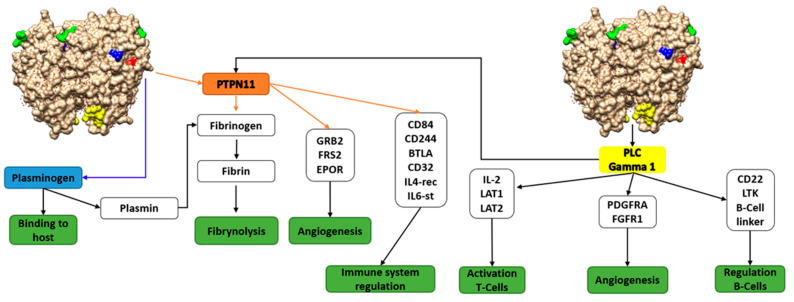
Interactions of the predicted virulence motifs. The motif colored in red interacts with PTPN11 and the one in yellow interacts with PLC gamma 1. As can be seen, both proteins are involved in tissue regeneration and immunomodulation. PTPN11 (protein tyrosine phosphatase non-receptor type 11); PLC gamma 1 (phospholipase C-gamma 1). Green and blue motifs are those experimentally found according to the bibliography.

**Table 1 microorganisms-09-01300-t001:** Plasminogen binding motifs in *Streptococcus pneumoniae* Enolase.

Sequence	Function	Distribution
FYDKERKVY	Binding to plasminogen [39]	Only Streptococcus
KK	Binding to plasminogen, according to published papers [40]	Pathogenic and non-pathogenic species
KxxK	Binding to plasminogen, according to published papers [40]	Pathogenic and non-pathogenic species
Y[LIV]E[LIV] Y[LIV]ED[PLIV]	Binding to PLCgamma –> Blood coagulation and interactionwith PTPN11 (Predicted)	Mostly in pathogenic species of different evolutive pathways
YTAV	Binding to PTPN11, a phosphatase related to blood coagulation and platelet formation (interaction with fibrinogen, fibrin,…). Noonan síndrome –> Inbalance in fibrynolitic components. (Predicted)	Mostly in pathogenic species of different evolutive pathways

**Table 2 microorganisms-09-01300-t002:** Some of the GO codes shared between metastasis and virulence-related moonlighting proteins. The abundances compared to the human proteome and human moonlighting proteins are shown in each case.

GO Number	Function	% Human Proteome	% Moonlighting Proteins	*p*-Value
001968	Binding to Fibronectin	0.39	3.06	6.87 × 10^−^^34^
0005518	Binding to Collagen	0.22	1.96	1.99 × 10^−^^26^
0050840	Binding to ECM	0.23	1.89	2.20 × 10^−^^16^
0043236	Binding to Laminin	0.15	1.32	3.01 × 10^−^^17^
0002020	Protease activity	0.72	5.12	2.20 × 10^−^^16^

## Data Availability

The data presented in this study are available in Appendix A.

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
