# Peer review of "Pathogen Moonlighting Proteins: From Ancestral Key Metabolic Enzymes to Virulence Factors"

_microorganisms, 2021, doi:10.3390/microorganisms9061300_

Round 1

Reviewer 1 Report

The authors tried to describe the virulence activity of moonlighting proteins based on in silico analysis. They suggested a common therapeutic target for cancer and infectious diseases. My comments are listed below.

The authors focused on pathogens in general. However, it would come with a better result if they focus on a single pathogen and host interaction.

If possible I would suggest the authors add data on mechanisms to inhibit moonlighting proteins to reduce their virulence including adhesion and invasion.

Line 124-129: In the result section, instead of focusing only on the current data, the authors tried to justify most of their previous work data. 

Line 141-163: this paragraph contains opposite sentences and brings confusion to the reader. It indicates that Moonlighting proteins are not a good candidate for the vaccine. However, in another sentence they indicated that these proteins are suitable for vaccine design developments, they do not have sequence homology with the host and no significant similarity is found with the human proteome. Hence, the authors need to revise the sentence and clear their justification.

Furthermore, the authors mostly depend on theoretical output to justify their results instead of depending on their data they obtained from this study.

Reviewer 2 Report

In the article “Pathogen moonlighting proteins: From ancestral key metabolic enzymes to virulence factors” the authors’ summarized computer analyzes that may be useful for understanding moonlighting proteins phenomenon.

There remain some issues that authors should consider revising this article.

After reading the paper, I do not fully understand what the authors mean in question 2. In my opinion, the question should be rewritten.

The authors mentioned that “several motifs that might be important in the adhesion to the ECM were identified”. After reading the entire work, I have the impression that the authors compared the obtained results with only one experimental work on the yeast of the genus Candida. This is a valuable comparison, but there is miss information on comparing the fragments identified in the computer analyzes of bacterial GAPDH or enolase-interacting with host proteins with the results from experimental work. Examples, for Gapdh S. pneumoniae two binding sites for human plasminogen (aa 113-132 and aa 303-322) were identified, for Gapdh M. pneumoniae one fragment (aa 326-337) involved in the binding of host proteins was identified – plasminogen, fibrinogen, fibronectin, and vitronectin (DOI: 10.1074/jbc.M116.764209; DOI: 10.1128/mSphere.01027-20).

Line 44: However, they are secreted outside the bacterial cell by an alternative mechanism and there they interact with host extracellular matrix (ECM) proteins [9]. In this sentence, do the authors suggest that only the protein secreted outside the cell interacts with the ECM? Some reports suggest that the protein displayed on the surface may also interact with the ECM, meaning it acts as an adhesin.

Line 136: What do you mean by unique virulence secreted factor. Can you expand it?

Line 151: This work focuses on bacterial proteins. I am not convinced that the reference to a viral protein is a good solution. For me, it would be more interesting to add an example of a bacterial protein used/or planned as a vaccine candidate and to make such a comparison for it.

Line 160: This hypothesis suggests an important idea usable in recombinant vaccine design: it is not recommended the use of proteins whose amino acids sequence share stretches between the pathogen and the host to design a new vaccine. Do the authors believe that moonlighting protein vaccines should not be used? Maybe this should be looked at from the other side. Namely, vaccines should be designed so that they are based on epitopes that differ from the host. This approach could make sense because moonlighting proteins are abundant on the pathogenic surface. What do the authors think about this?

In line 385 the authors claim that plasminogen is a component of ECM. I cannot agree with that. Improvement is needed.

Some additional minor comments:

Fig.2 It is a good scientific practice to give a PDB ID in text and figure legends.

In several sentences (for example lines 112, 176, 182), plasminogen is spelled with its full name even though an abbreviation was introduced previously. Verify this throughout the text.

Line 181: For example, macrophages recruit surface GAPDH where it also functions as a plasminogen receptor. The plasminogen binding of GAPDH allows it to digest the extracellular matrix, thus facilitating macrophage migration. The reference is missing

In my opinion, table 1 should contain citations.

The abbreviation PTPN11 and PLC should be clarified.

Moreover, the abbreviation should be standardized. PLC gamma appears in the text (line 217) and PLC gamma1 appears in figure 3.

Line 252: GAPDH of Mycoplasma suis has two pig partners, actin and a structural protein (Supplementary Information S11). Are you sure that you find pig proteins?

Round 2

Reviewer 2 Report

I accepted your answers.